# Imaging of Bioprosthetic Valve Dysfunction after Transcatheter Aortic Valve Implantation

**DOI:** 10.3390/diagnostics13111908

**Published:** 2023-05-29

**Authors:** Louhai Alwan, Benedikt Bernhard, Nicolas Brugger, Stefano F. de Marchi, Fabien Praz, Stephan Windecker, Thomas Pilgrim, Christoph Gräni

**Affiliations:** Department of Cardiology, Inselspital, University of Bern, 3010 Bern, Switzerland

**Keywords:** TAVI, multimodality imaging, bioprosthetic valve dysfunction, bioprosthetic valve failure, structural valve deterioration

## Abstract

Transcatheter aortic valve implantation (TAVI) has become the standard of care in elderly high-risk patients with symptomatic severe aortic stenosis. Recently, TAVI has been increasingly performed in younger-, intermediate- and lower-risk populations, which underlines the need to investigate the long-term durability of bioprosthetic aortic valves. However, diagnosing bioprosthetic valve dysfunction after TAVI is challenging and only limited evidence-based criteria exist to guide therapy. Bioprosthetic valve dysfunction encompasses structural valve deterioration (SVD) resulting from degenerative changes in the valve structure and function, non-SVD resulting from intrinsic paravalvular regurgitation or patient–prosthesis mismatch, valve thrombosis, and infective endocarditis. Overlapping phenotypes, confluent pathologies, and their shared end-stage bioprosthetic valve failure complicate the differentiation of these entities. In this review, we focus on the contemporary and future roles, advantages, and limitations of imaging modalities such as echocardiography, cardiac computed tomography angiography, cardiac magnetic resonance imaging, and positron emission tomography to monitor the integrity of transcatheter heart valves.

## 1. Introduction

Based on the favorable outcomes of recent randomized clinical trials, transcatheter aortic valve implantation (TAVI) is today not only conducted in high- and intermediate-risk populations, but also increasingly performed in younger-, intermediate- and lower-risk patients with severe symptomatic aortic stenosis [1,2,3,4,5,6]. With expanded use in populations with a longer life expectancy, bioprosthetic valve dysfunction and its terminal stage bioprosthetic valve failure (BVF) are expected to become a major cause of cardiovascular morbidity, underlining the need to investigate the long-term durability of transcatheter heart valves (THV). Although the durability of THV up to 5 years has already been demonstrated, with low rates of BVF shown in landmark trials, data on clinical long-term outcomes and THV integrity after 5 years are still scarce [7]. The European Association of Percutaneous Cardiovascular Interventions (EAPCI), the European Society of Cardiology (ESC), and the European Association for Cardio-Thoracic Surgery (EACTS) introduced in 2017 standardized criteria to define bioprosthetic valve dysfunction that aim at standardization in the data reporting of future studies assessing the long-term durability of THV [8]. The consensus statement classifies bioprosthetic valve dysfunction into structural valve deterioration (SVD), non-structural valve deterioration (non-SVD), valve thrombosis, endocarditis, and their common end stage BVF (Figure 1). To detect and differentiate these entities and monitor valve function over time, the validation of imaging modalities is an important requirement. In this narrative review, we will elucidate the contemporary role, advantages, and limitations of transthoracic (TTE) and transesophageal echocardiography (TEE), cardiac computed tomography angiography (CCTA), cardiac magnetic resonance imaging (CMR), and positron emission tomography (PET) in monitoring anatomical and functional THV integrity and their potential impacts on patient management.

## 2. Structural Valve Deterioration (SVD)

SVD is commonly defined as permanent changes in the THV structure due to leaflet thickening, calcification, tears and disruption, or pannus formation (morphological SVD) that can result in obstruction (stenosis) and/or intra-prosthetic regurgitation (hemodynamic SVD) [8,9,10,11]. Based on the hemodynamic severity of stenosis or regurgitation, the red flags and stages of SVD are classified according to Table 1 and Table 2, following the definitions of the European Society of Cardiology and the Valve-in-Valve International Data (VIVID) [8,10]. The widespread accessibility of echocardiography and its capacity for a direct comparison between pre- and post-interventional and follow-up imaging findings have established TTE as the primary imaging modality for the monitoring of valve function and for the timely detection of SVD.

At the early stages, structural changes without deterioration in valve hemodynamic function are observed (VIVID Stage 1) and can progress to Stage 2 with moderate hemodynamic valve deterioration and finally to Stage 3 with obstruction (stenosis) or regurgitation and BVF. The finding of an increased mean transvalvular gradient (≥20 mm Hg) and/or a small valve effective orifice area (EOA) (<1.1 cm^2^), a low Doppler velocity index (DVI) (<0.35), an acceleration time/left ventricular (LV) ejection time ratio >0.32, or the new onset or worsening of transprosthetic regurgitation is a red flag for SVD [12]. However, the specificity of these individual findings is limited, and elevated transprosthetic gradients may be caused by a prosthesis–patient mismatch (PPM) in the absence of valve obstruction. An improvement in cardiac output can increase the transvalvular gradients at follow-up and should not be mistaken for prosthetic valve obstruction. Stable or increased EOA and Doppler velocity indexes are further parameters that can help to differentiate the two entities. Reintervention should not be considered only on the basis of a high gradient and/or a small aortic valve area (AVA) or DVI at a single TTE during follow-up unless the gradient is extremely high (50 mm Hg) and valve anatomy is obviously pathological.

If a higher image resolution is required or if the echocardiographic transthoracic windows are inadequate for quantitative/detailed assessment, TEE is the optimal alternative to accurately assess the valve morphology and function [13]. TEE enables the evaluation of leaflet mobility, as well as the detection of additional anomalies compatible with pannus, thrombus, or non-infection and infection vegetations, keeping in mind that echocardiography does not permit tissue differentiation between these entities and that the diagnosis should take into account both the clinical presentation and the localization, size, aspect, and mobility of the lesion. However, the assessment of a prosthetic aortic valve by TEE may be more challenging. Both esophageal and transgastric views are necessary to better characterize the leaflet and sub-valvular morphology. A deep transgastric view mimicking an apical “5-chamber” view allows the assessment of the gradient across the prosthesis and a better evaluation of the leaflet motion, prosthetic aortic regurgitation, and sub-valvular morphology. Although non-contrast CCTA can be used to identify the presence of leaflet calcifications, an early sign of THV degeneration, four-dimensional (4D) CCTA (including contrast) is the preferred and most valuable adjunctive modality to assess not only the valve structure but also leaflet motion and indirect signs of obstruction (stenosis) or regurgitation (Table 2) [14]. CCTA should be seen as an additional modality to echocardiography (Table 3), as it only allows one to assess the anatomy and cannot provide information on the transvalvular gradients or regurgitation fraction. Further, CMR might be used in patients with a poor echocardiographic window to derive the EOA, which has been shown to correlate to the EOA derived from echocardiography in both functional and dysfunctional THVs [10,13,15]; however, one has to be aware that metallic artifacts from THV can impair the CMR image quality.

PET is an emerging imaging modality in certain settings, capable of detecting the earlier stages of SVD via the visualization of “inflammation” using different tracers [16,17,18]. However, the ways in which these advanced imaging modalities should be used in addition to TTE/TEE in SVD, and which findings indicate the need for reintervention, are currently under discussion.

## 3. Non-Structural Valve Deterioration (Non-SVD)

Non-SVD is bioprosthetic valve dysfunction due to extrinsic factors such as paravalvular regurgitation (PVR) (Figure 2), PPM, device malpositioning (including procedural malpositioning and valve migration), under- or oversizing, abnormal frame expansion, or sub-valvular pannus overgrowth. Although the presence of non-SVD is not a deterioration in the THV itself, it might result in the early development of SVD [8,10,19] and can often no longer be differentiated from SVD after disease progression. In the majority, non-SVD is directly related to the initial valve replacement intervention, is already present immediately after intervention, and often remains stable throughout follow-up [8]. However, in a few cases, non-SVD may improve or worsen at follow-up and can result in BVF and the need for reintervention.

### 3.1. Paravalvular Regurgitation (PVR)

PVR can result from the excessive or asymmetric distribution of calcification, or from THV malpositioning. Degenerative calcification of the native aortic valve, a high or low position of the prosthesis within the aortic root, undersizing of the valve, and a bicuspid aortic valve are risk factors for the development of PVR [20]. With the newer-generation THVs—with an additional sealing skirt around the valve prosthesis stent frame—PVR rates have been reduced to a range of 1–3% for moderate PVR and 29–36% for mild PVR in the latest low-risk TAVI trials [2,3]. The impact of mild PVR on LV function, symptoms, and long-term mortality in lower-risk patients with a longer life expectancy is still unknown. However, data from the PARTNER-1 trial have suggested decreased survival in this subset [21], underlining the need for early diagnosis.

Immediately after TAVI, aortic root angiography can be used to screen for PVR and can guide corrective actions (post-dilatation or valve-in-valve implantation) [22]. According to the Valve Academic Research Consortium 2 (VARC 2) criteria, PVR can be angiographically categorized into three degrees: mild (reflow of contrast in the outflow tract and middle portion of the LV but clearing with each beat), moderate (reflow of contrast in the entire left ventricular cavity with incomplete washout in a single beat and faint opacification of the entire LV over several cardiac cycles), and severe (opacification of the entire LV with the same intensity as in the aorta and persistence of the contrast after a single beat) [22]. Additionally, the aortic regurgitation index (ARI) has been proposed for the hemodynamic assessment of the PVL and can be calculated as ARI = [(diastolic blood pressure − LV end diastolic pressure)/systolic blood pressure]*100 [22].ARI has been found to inversely correlate to PVR and is associated with 1-year mortality; ARI < 25% is associated with an increased 1-year mortality risk in comparison with patients with ARI > 25% [22]. In order to evaluate PVL after TAVI and to identify patients who will benefit from corrective measures (post-dilatation, valve-in-valve implantation), Sinning et al. have suggested using a multimodal approach combining hemodynamic measurements and imaging modalities (aortography and TTE/TEE) [22]. When there is no PVL and/or an ARI > 25% present, no additional measures need to be taken. Meanwhile, in patients with more than a mild PVL and/or ARI < 25%, the evaluation of PVL using TTE/TEE is recommended to elucidate the cause of PVL.

After TAVI, Doppler echocardiography is the method of choice for the quantitative and semi-quantitative evaluation of both central and paravalvular jets [23,24,25]. The VARC 2 recommendations advocate for an echocardiographic evaluation before hospital discharge to establish baseline parameters. Color Doppler evaluation should focus on the left ventricular outflow tract (LVOT) extremity of the THV for PVR, and for central regurgitation at the point of leaflet coaptation. PVR most frequently occurs at the location of the native valve commissures, because, at these locations, the THV, which has a circular shape with limited flexibility, does not conform to the triangular configuration of the commissure [26,27]. To distinguish between moderate and severe PVR, Doppler measurements may be used, particularly the flow reversal detected by pulsed-wave Doppler in the descending aorta, which, however, can be distorted in patients with noncompliant aorta and/or LV diastolic dysfunction [25,28]. Quantitative parameters such as the prosthetic regurgitant volume, effective regurgitant orifice area, and regurgitant fraction can be assessed to further characterize PVR. However, because PVR jets are frequently multiple, irregular, and eccentric, it can be difficult to detect and grade PVR using echocardiography [25,28]. TEE is a valid alternative to TTE if a higher spatial resolution is required and can precisely evaluate the valve structure and hemodynamic parameters. Moreover, TEE may be helpful to differentiate paravalvular regurgitation (non-SVD) from transvalvular regurgitation (structural SVD). Of note, transvalvular regurgitation can also occur after aggressive postdilatation and should be considered in the differential diagnosis. Three-dimensional (3D) echocardiography may eventually overcome the limitations of two-dimensional (2D) and standard Doppler measurements in quantifying PVR. Studies using 3D TTE have shown the feasibility of measuring the 3D vena contracta of PVR following TAVR [29,30]. Moreover, 3D TEE can provide a more detailed assessment of the aortic regurgitation and aortic valve morphology and function; 3D echocardiography can reliably identify the mechanism of aortic regurgitation and, with the complementary use of 3D color mode, provides important diagnostic information about the regurgitant jets’ size, location, and severity. Newer approaches based on real-time 3D velocity color flow Doppler echocardiography allow the automated quantification of the velocity, flow rate, and flow volume in any given region of the heart from color Doppler images [31]. This feasible method provides excellent accuracy and reproducibility for the quantitation of aortic stroke volumes. A more precise assessment of the PVR regurgitant volume may be obtained by using this technique to measure the LV and RV stroke volumes.

Direct quantification of PVR is most accurate using CMR via the 2D phase-contrast velocity approach. The antegrade and retrograde aortic flows are measured using phase-contrast through plane imaging in a short-axis plane that cuts the aorta slightly above the THV. The regurgitant volume is calculated as the integral of the retrograde flow over time, the total forward volume as the integral of the anterograde aortic flow over time, and the regurgitant fraction as follows [32]: (regurgitant volume/total forward volume) × 100. The advantages of CMR over echocardiography include its excellent reproducibility and diagnostic accuracy, which are less affected by multiple and/or eccentric PVR jets [30]. Limitations are the possible lack of accessibility of scanners, arrhythmia that lowers the measurement accuracy, patient movement resulting in motion artifacts and lowering the quality of image acquisition, flow turbulence, a loss of signal near the THV stent (artifacts), and a slight overestimation of PVR due to the coronary artery diastolic flow included in the final regurgitant volume assessment [33]. To overcome the issue of stent artifacts, a more distal assessment of the flow in the aorta ascendens with regard to the aortic valve or alternatively within the aorta descendens may improve the image quality. The lack of specific cutoffs for the grading of PVR severity using regurgitation fractions from CMR remains an issue. In a head-to-head comparison of echocardiography vs. CMR, CMR reclassified the paravalvular regurgitation severity in approximately 35–40% of patients with at least one grade higher [34,35,36].

### 3.2. Patient–Prosthesis Mismatch (PPM)

PPM is a condition in which the EOA of a normally functioning THV is too small in relation to the patient’s body surface. PPM is classified as severe if the indexed EOA is less than 0.65 cm^2^/m^2^, as moderate if between 0.65 and 0.85 cm^2^/m^2^, and as not hemodynamically relevant if greater than 0.85 cm^2^/m^2^. In obese patients (body mass index > 30 kg/m^2^), lower cutoff values of the indexed EOA (i.e., ≤0.70 and ≤0.55 cm^2^/m^2^ for moderate and severe PPM, respectively) should be applied to identify aortic PPM [25].

The identification and quantitation of PPM are primarily based on Doppler TTE [37]. PPM should be suspected when a high peak aortic velocity and mean transprosthetic gradient is found without signs of SVD. Then, the EOA should be obtained by the continuity equation from the LVOT area. Since the measurement of the LVOT area by TTE is of limited reproducibility, the EOA can also be obtained by fusion or hybrid imaging with CCTA-based measurement of the LVOT and the velocities measured by Doppler [37].

Notably, Doppler flow-dependent parameters alone are of limited utility to differentiate between PPM and degenerative valve obstruction, since valve obstruction presents similar hemodynamic characteristics. To differentiate the two entities, VARC 2 suggests the determination of one flow-dependent criterion (such as the mean gradient) and one flow-independent criterion (such as the EOA) as a first step. The calculated EOA should be compared with normal values derived from in vivo studies and, when larger than the “reference value -1SD”, it should be considered normal. If the mean gradient is high and the EOA “normal”, the Doppler velocity index (DVI) should be determined. The DVI is the ratio of the velocity time integral of the LVOT and of the bioprosthesis. The DVI is frequently >0.30–0.35 in patients with isolated PPM, whereas it is <0.35 in patients with possible obstruction and <0.25 in patients with significant obstruction (Figure 3). A normal DVI suggests a well-functioning THV, and the indexed EOA can be utilized to ascertain the cause of the initial discordance [25]. Additionally, imaging of the valve leaflet morphology and mobility is essential to differentiate PPM from prosthetic obstruction. While the bioprosthetic valve obstruction results from leaflet thickening and decreased mobility, PPM features a normal valve structure, leaflet morphology, and mobility. A leaflet thickness greater than 2 mm is considered abnormal. Moreover, low-dose dobutamine stress echocardiography allows for differentiation between a true prosthetic malfunction and mismatch in patients with a low valvular flow, low resting effective orifice area (EOA), and decreased DVI.

Other imaging modalities, including CCTA, may offer complementary information in the evaluation of PPM [37]. CCTA is helpful in identifying and assessing the mobility and morphology of the leaflet and can also assess the valve thrombus and differentiate thrombus versus pannus. In CCTA, PPM is characterized by a normal EOA, small indexed EOA, and normal leaflet mobility and morphology.

## 4. Valve Thrombosis/HALT/Pannus

Valve thrombosis may be classified into subclinical thrombosis and clinically significant thrombosis. Clinically significant valve thrombosis after TAVI typically presents with an increase in transvalvular gradients and symptoms of heart failure. On the other hand, subclinical leaflet thrombosis is an incidental finding on 4D CCTA or TEE imaging, which does not cause symptoms but can present with elevated transvalvular pressure gradients. The recommendations of the American College of Cardiology incorporate several functional criteria to define thrombosis-associated subclinical valve stenosis, such as the peak prosthetic aortic jet velocity, mean gradient, and EOA. Subclinical stenosis may be present in patients with a peak prosthetic aortic jet velocity of 3–4 m/s, a mean transvalvular gradient of 20–35 mmHg, and an EOA between 0.88 and 1.2 cm^2^/m^2^, while a peak prosthetic aortic jet velocity > 4 m/s, mean gradient > 35 mmHg, and EOA < 0.8 cm^2^/m^2^ are suggestive of significant stenosis. The European recommendations further consider an increase in transvalvular mean gradient during stress echocardiography or an increase at follow-up between 10 and 19 mmHg compared to baseline as a sign of subclinical valve thombosis, while an increase ≥ 20 mmHg at follow-up implicates thrombosis-associated obstruction [8,12]. Further testing, using 4D CCTA or TEE, should be carried out to confirm the diagnosis and verify the reduced leaflet motion. However, CCTA is the imaging modality of choice for THV subclinical and clinical thrombosis, evaluating both leaflet mobility and thickness (Figure 4 and Figure 5) [38].

More recently, an anatomical finding on THVs from CCTA, the so-called hypoattenuated leaflet thickening (HALT), was discussed in the setting of valve thrombosis [39,40,41,42,43,44] and defined as a visually identified increased leaflet thickness with the typical meniscal appearance on long-axis view [45]. HALT is visualized in CCTA as hypoattenuated lesions that are typically seen at the periphery and base of the leaflet on longitudinal imaging and appear as hypodense lines. On axial views with 3D volume-rendered imaging, leaflets appear as wedge- or crescent-shaped opacities in both systole and diastole. HALT can cause restricted leaflet movement (RLM), or hypoattenuated leaflet motion (HAM), usually without severely elevated transvalvular gradients on echocardiography. The assessment of RLM is based on the maximal degree of leaflet opening in the systolic phase. Leaflet immobility is graded as normal (no RLM); mildly (<50% RLM), moderately (50–70% RLM), or severely (>70% RLM) impaired; or immobile (100% RLM) [46]. The association between HALT and RLM is not fully understood and not all thickened leaflets show RLM, suggesting that these findings are two stages of the same phenomenon, with leaflet thickening occurring earlier, followed by RLM at a more advanced stage. On CCTA, the definition of clinically relevant leaflet thrombosis currently requires the presence of both HALT and RLM [46], in addition to the presence of clinical sequelae of a thromboembolic event or worsening obstruction or regurgitation. In the absence of clinical sequelae, severe hemodynamic valve deterioration must be present in addition to HALT and RLM. Subclinical leaflet thrombosis is considered when imaging findings of HALT are present without or only mild valve hemodynamic deterioration and no symptoms/sequelae. Subclinical thrombosis may resolve spontaneously without any treatment in up to 50% of cases [45], and, at the present time, there is no evidence that subclinical leaflet thrombosis requires anticoagulation therapy, since no significant impact on clinical outcomes is observed. Clinically relevant valve thrombosis, on the other hand, requires therapy with a vitamin K antagonist. Even with treatment, it may evolve to valve leaflet fibrosis and calcification and thus become irreversible (i.e., structural SVD), eventually leading to BVF and reintervention.

The incidence of valve thrombosis is much higher when CCTA anatomical criteria such as HALT are considered, as compared to echocardiographic functional criteria alone. In two retrospective analyses, the prevalence of clinically relevant valve thrombosis was reported to be 0.6 and 2.8% after TAVI, whereas the prevalence of subclinical leaflet thrombosis has been reported to be as high as 15–35% in studies assessing this phenomenon by means of TEE and/or CCTA [39,40,41,42,44]. Accordingly, it has been suggested that CCTA may provide the highest sensitivity to detect THV thrombus, especially at the early stages [47,48]. CCTA is also clinically useful to distinguish between thrombus and pannus [12] and is therefore the imaging modality of choice for suspected THV subclinical and clinical thrombosis, evaluating both leaflet mobility and thickness [38].

## 5. Infective Endocarditis

Another potentially reversible mechanism of bioprosthetic valve dysfunction is infective endocarditis. Bioprosthetic valve endocarditis is defined by at least one of the following criteria: (i) fulfillment of the modified Duke criteria for “definite infective endocarditis” (two major, one major, and three minor or five minor criteria); (ii) evidence of vegetation, abscess, or pus confirmed as secondary to infection by histological or microbiological studies during reoperation; and (iii) evidence of abscess, pus, or vegetation confirmed on autopsy. Endocarditis frequently results in hemodynamic and morphological valve degeneration, which might result in stage 2 or 3 SVD (obstruction and/or regurgitation).

Echocardiography remains the first-line modality in suspected infective endocarditis. Due to prosthetic shadowing, TTE can yield negative results in THV endocarditis and should be therefore followed by TEE. Even in case of negative TEE, the procedure should be repeated due to the high clinical likelihood of infective endocarditis [49,50]. TEE provides high accuracy for the detection and measurement of vegetations, both of which have a significant impact on the risk of embolism and the recommendation for early surgery; it is also useful for the detection and follow-up of paravalvular abscesses.

CCTA and PET-CT may offer important information in patients with poor echocardiographic image quality or in unclear cases [51,52]. In comparison to TEE, CCTA has fewer imaging artifacts due to the prosthetic valve and enables the identification of paravalvular complications such abscesses or aneurysms [53,54]. However, it is less sensitive than TEE in detecting small vegetations.

For the detection of peripheral emboli and cardiac or extracardiac infection sites, radiolabeled leukocyte scintigraphy or 18-F-fluorodeoxyglucose positron emission tomography–computed tomography (FDG-PET/CT) scanning may be useful. According to most recent recommendations, reclassifying potential diagnoses as definite infective endocarditis by adding FDG-PET/CT findings as an additional significant criterion to the modified Duke criteria enhances the sensitivity without compromising the specificity [49,50].

## 6. Bioprosthetic Valve Failure (BVF)

BVF describes the end stage of the above-mentioned entities and the clinical consequences, such as progressive heart failure, high mortality, and the requirement for reintervention [10]. According to the VARC 3 and EACPI/European Association of Cardio-Thoracic Surgery criteria [8,10,55], BVF can be diagnosed in the presence of severe bioprosthetic valve dysfunction (SVD, non-SVD, thrombosis, or endocarditis) associated with clinically suggestive criteria (i.e., new-onset or worsening symptoms, LV dilation/hypertrophy/dysfunction, or pulmonary hypertension) or irreversible Stage 3 bioprosthetic valve dysfunction in the absence of clinically expressive criteria.

## 7. Future Perspectives on the Early Diagnosis of Bioprosthetic Valve Degeneration

Despite refinements in established imaging modalities and studies on parameters that might refine the diagnosis and risk stratification, the evaluation of THV remains challenging. Therefore, further modalities such as 18F-NaF PET may aid in the evaluation of bioprosthetic valve degeneration. The development of biomarkers and efficient medical treatments is delayed due to the poor understanding of the pathophysiology of AS. Since calcification and inflammation are expected to play a significant pathogenic role, non-invasive markers of their activity are important in better understanding the cause of this condition and in predicting disease progression.

Recent studies have investigated 18F-NaF PET as a marker of vascular calcification in AS and atherosclerosis affecting the aorta and coronary and carotid arteries [18,56,57,58,59]. In bone, 18F-NaF is thought to be incorporated onto the surface of hydroxyapatite crystal [60]. Given that hydroxyapatite is also a key component of vascular calcification, it too has been the presumed radiotracer target in AS and atherosclerosis. The autoradiography and immunohistochemistry data from Dweck et al. showed a strong association between 18F-NaF activity and osteocalcin staining, a well-known osteogenic protein that binds to hydroxyapatite.

The results of 18F-NaF PET of THVs are a powerful independent predictor of subsequent hemodynamic bioprosthetic valve degeneration that is applicable to TAVI. Valvular 18F-NaF uptake offers an evaluation of disease activity and a prediction of subsequent disease progression and clinical outcomes in patients with aortic stenosis [18,61]. Additionally, Kwiecinski et al. showed that 18F-NaF uptake still occurs in the original aortic valve that is retained in all TAVI patients [62]. This discovery was corroborated by ex vivo studies that showed histological evidence of continued calcification activity in native aortic valve tissue several years after TAVI. Native aortic valve 18F-NaF uptake and calcification activity were high [18,61,63,64,65]. The fact that it continues for several years after TAVI, when mechanical stresses are no longer being exerted on the valve leaflets, confirms that aortic stenosis is an active, regulated disease process and not simply the result of valve wear and tear. Increased 18F-NaF uptake in the bioprosthetic valve leaflets has been shown by Kwiecinski et al. to be a more accurate predictor of later valve malfunction than the valve age, cardiovascular comorbidities, and imaging evaluations from echocardiography and CCTA. Since other imaging modalities such as echocardiography and CT are currently limited in this area, 18F-NaF PET may be a pioneer in the prediction of bioprosthesis failure. It is a highly promising marker of early bioprosthesis degeneration.

Molecular PET has been investigated to understand the pathomechanisms behind the deterioration of bioprostheses. Within valve leaflets, 18F-NaF PET detects calcification activity and serves as an early indicator of valve degeneration. In two recent multicenter observational studies involving patients with both surgical and transcatheter bioprostheses, 18F-NaF PET was the most effective predictor of a subsequent deterioration in valve hemodynamics and the emergence of overt valve failure. It detected evidence of valve degeneration that was not visible on echocardiography or contrast-enhanced CT [62,63]. PET-CT is particularly useful when prosthetic valve endocarditis is suspected but TTE and TEE imaging are normal.

In order to assist physicians during TAVI procedures, a new, entirely automatic method combining the real-time fusion of 3D TTE and CCTA images on live fluoroscopy was developed [66]. With this novel approach, the implantation site may be seen in live images more precisely, without the need for contrast material. Accurate valve deployment is made easier by the improved intra-procedural resolution, which also improves the valve’s position with regard to the aortic annulus. High-resolution CCTA imaging may also be helpful in the diagnosis and monitoring of subclinical leaflet thrombosis. As already indicated, CCTA shows the capacity to detect subclinical leaflet thrombosis in a considerable proportion of patients, but its routine usage is presently not advised due to the uncertain clinical implications of these results.

CMR provides an alternative to echocardiography in assessing valvular hemodynamics in patients with BPHVs and has been shown to be relatively safe for imaging at both 1.5 T and 3.0 T for a number of valve types. When echocardiography is constrained by imaging windows, CMR may be a helpful alternative method. In fact, it has been demonstrated that in both normal and dysfunctional BPHVs, the effective orifice area estimated from MRI correlates with the effective orifice area estimated from echocardiography [67,68,69].

## 8. Conclusions

The timely identification of dysfunction in bioprosthetic valves is critical for patients undergoing TAVI. Echocardiography remains a first-line, low-cost, non-invasive imaging method for the evaluation and monitoring of THV function due to its ability to provide immediate, clinically relevant information.

Multimodality imaging is leading the way in detecting the earlier stages of valve degeneration, allowing personalized and optimal management strategies for patients with THVs. Further, imaging provides a window into the pathophysiology and can determine the mechanisms of degeneration, which will ultimately lead to improved valve durability and improved outcomes.

## Figures and Tables

**Figure 1 diagnostics-13-01908-f001:**
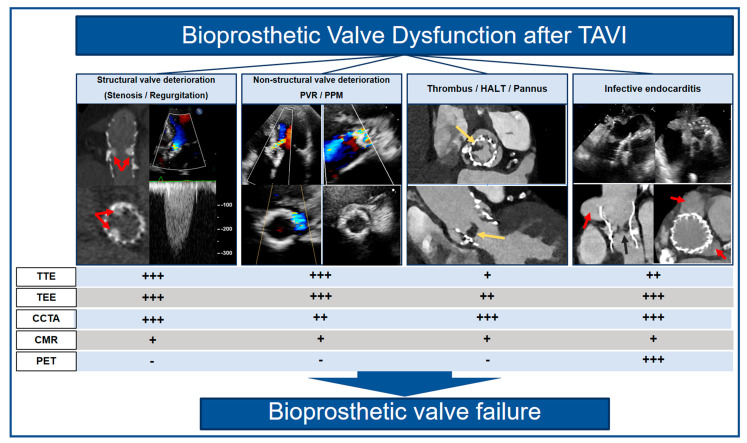
Multimodality imaging can be used to depict underlying causes (i.e., structural valve deterioration marked with red arrows on the left panel, non-structural valve deterioration, thrombus marked with yellow arrows, hypoattenuated leaflet thickening, pannus and infective endocarditis marked with red arrows on the right panel) of bioprosthetic valve dysfunction after TAVI. These changes may ultimately lead to bioprosthetic valve failure.

**Figure 2 diagnostics-13-01908-f002:**
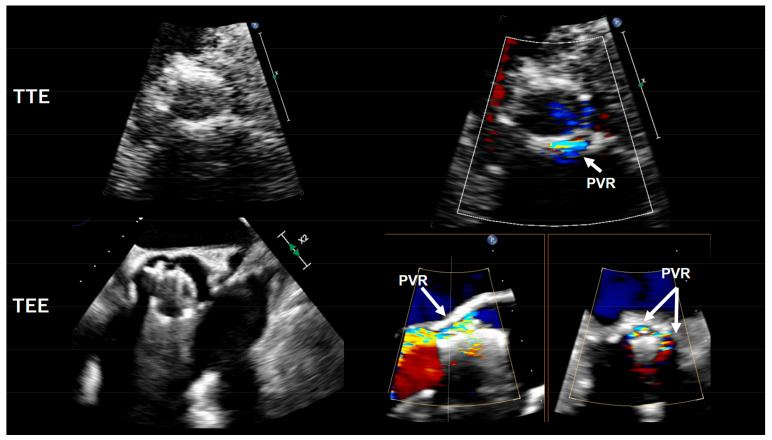
Demonstration of paravalvular leak (PVR) after TAVI by transthoracic echocardiography (TTE) and transesophageal echocardiography (TEE).

**Figure 3 diagnostics-13-01908-f003:**
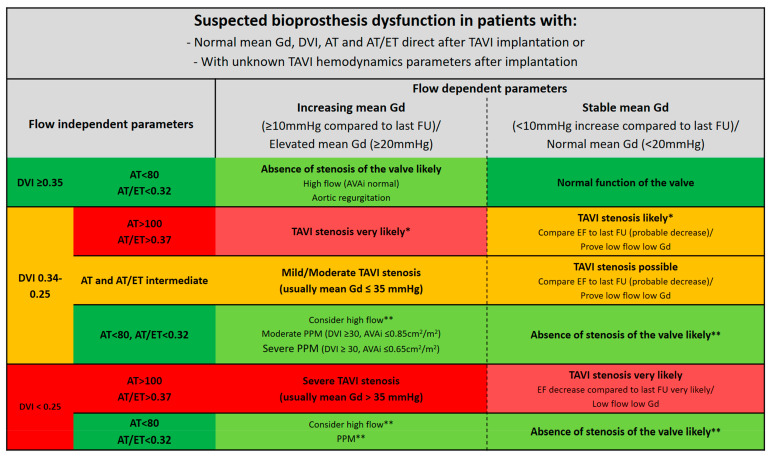
Different clinical scenarios of suspected bioprosthesis dysfunction after TAVI. Adapted from Pibarot P. et al., 2019 [37]. Gd = gradient; DVI = Doppler velocity index; AT = acceleration time; AT/ET = quotient acceleration time on ejection time; EF = ejection fraction; FU = follow-up; VTI = velocity time integral. * Improper high LVOT VTI (septum bulge, measurement done in the valve)/underestimation of TAVI gradient due to improper CW–Doppler placement. ** Improper low LVOT VTI (measurement done in LV/big LVOT).

**Figure 4 diagnostics-13-01908-f004:**
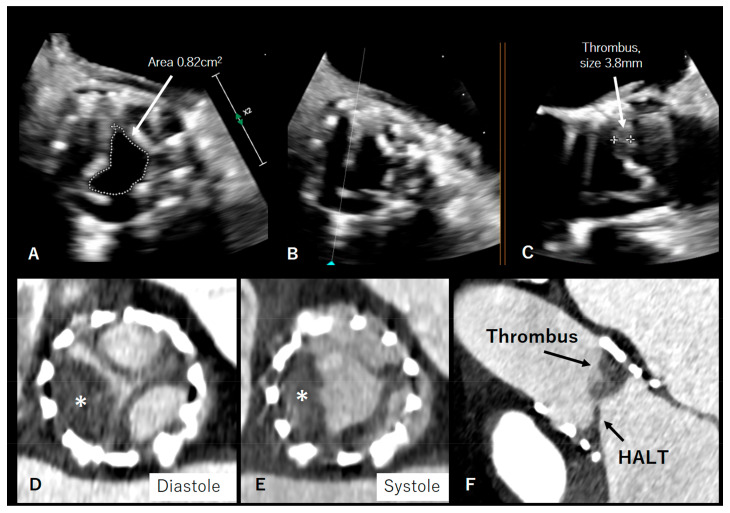
In panel (**A**–**C**), echocardiographic findings of leaflet thickening and thrombosis after TAVI 29 mm Edwards Sapien are depicted. Panel (**D**–**F**) shows hypoattenuated leaflet thickening (HALT) and valve thrombosis (asterisk).

**Figure 5 diagnostics-13-01908-f005:**
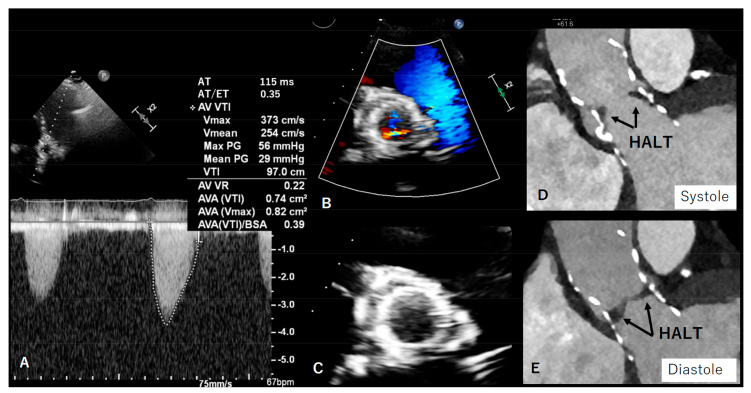
In panel (**A**–**C**), transthoracic echocardiography images are depicted and show increased peak velocity, mean gradient, and acceleration time and reduced Doppler velocity index, without obvious leaflet thickening (due to impaired image quality, impaired echocardiographic window). Panel (**D**,**E**) shows hypoattenuated leaflet thickening (HALT) with impaired motion as the underlying cause of increased gradient on the TAVI bioprosthesis and is therefore considered subclinical leaflet thrombosis.

**Table 1 diagnostics-13-01908-t001:** Red flags for bioprosthetic valve failure.

Red Flags
Color-flow Doppler systolic restriction
Leaflet abnormality (reduced mobility, thickening, calcification)
Mean gradient ≥ 20 mm Hg (≥30 mm Hg) *
Increase in mean gradient ≥ 10 mm Hg (≥20 mm Hg) * during follow-up
EOA < 1.1 cm^2^ (<0.8 cm^2^) *
DVI < 0.35 (<0.25) *
AT/LVET > 0.32 (>0.37) *
New onset or worsening of intraprosthetic AR ≥ mild

* Red flags with higher level of BVD suspicion. AR = aortic regurgitation; AT = acceleration time; DVI = Doppler velocity index; EOA = effective orifice area; LVET = left ventricular ejection time.

**Table 2 diagnostics-13-01908-t002:** Classification of structural valve deterioration.

Classification of Structural Valve Deterioration			
EAPCI/ESC/EACTS (Capodanno et al., 2017 [8])		VIVID (Dvir et al., 2018 [10])	
Class	Characteristics	Class	Characteristics
Not specifically defined		Stage 0	No significant change from immediate post-implantation Mean gradient < 20 mmHg Intravalvular regurgitation less than moderate (<2+/4+) No morphological leaflet abnormality, such as leaflet thickening
		Stage 1	Morphological leaflet abnormality without significant hemodynamic changes Leaflet calcification, sclerosis, thickening, or new leaflet motion disorder Absence of hemodynamic changes as defined in SVD Stage 0
Moderate hemodynamic SVD	Presence of any of the following: Mean transprosthetic gradient ≥ 20 mmHg and <40 mmHg Mean transprosthetic gradient ≥ 10 and <20 mmHg change from baseline Moderate intra-prosthetic aortic regurgitation, new or worsening (>1+/4+) from baseline	Stage 2S	Moderate stenosis Increase in transvalvular gradient of ≥10 mmHg and <20 mmHg with a concomitant decrease in EOA and DVI (0.25–0.35), which is not the result of isolated leaflet thickening Mean transprosthetic gradient ≥ 20 mmHg and <40 mmHg Peak velocity of 3–4 m/s
		Stage 2R	Moderate regurgitation Regurgitant fraction ≥ 10 and <30% Diastolic flow reversal in proximal descending aorta with end-diastolic velocity < 30 cm/s PHT < 500 and ≥200 ms Absence of a main paravalvular component
		Stage 2RS	Moderate stenosis AND moderate regurgitation Presence of characteristics of Stage 2S and 2R
Severe hemodynamic SVD	Presence of any of the following: Mean transprosthetic gradient ≥40 mmHg Mean transprosthetic gradient ≥20 mmHg change from baseline Severe intra-prosthetic aortic regurgitation, new or worsening (>2+/4+) from baseline, concomitant with decrease in EOA and DVI	Stage 3	Severe stenosis and/or severe regurgitation Increase in transvalvular gradient of ≥20 mmHg with a concomitant decrease in EOA and DVI (<0.25), which is not the result of isolated leaflet thickening Mean transprosthetic gradient ≥ 40 mmHg Peak velocity of >4 m/s Regurgitant fraction ≥ 30% Holo-diastolic flow reversal in proximal descending aorta with end-diastolic velocity ≥ 30 cm/s PHT < 200 ms Absence of a main paravalvular component
Morphological SVD	Presence of any of the following: Leaflet integrity abnormality (i.e., torn or flail causing intra-frame regurgitation) Leaflet structure abnormality (i.e., pathological thickening and/or calcification causing valvular obstruction or central regurgitation) Leaflet function abnormality (i.e., impaired mobility resulting in obstruction and/or central regurgitation) Strut/frame abnormality (i.e., fracture)	Not specifically defined, corresponds to class 1 to 3	

Left column (gray background) represent classification of structural valve deterioration according to EAPCI/ESC/EACTS (Capodanno et al., 2017 [8]), whereas the right column (green background) represent the classification according to VIVID (Dvir et al., 2018 [10]). DVI = Doppler velocity index; EOA = effective orifice area; SVD = structural valve deterioration; PHT = pressure half time.

**Table 3 diagnostics-13-01908-t003:** Comparison of cardiac imaging modalities in the assessment of aortic bioprosthesis valve dysfunction.

	SVD	Non-Structural BVD: PPM	Valve Thrombosis	Pannus	Valve Endocarditis
TTE/TEE	Diffuse or focal hyperechogenic leaflet thickening (>2 mm) Reduced leaflet mobility Paucity (restriction) of color Doppler transvalvular flow	Normal leaflet morphology and mobility Reduced indexed EOA	Diffuse or focal hypoechogenic leaflet thickening (>2 mm) Normal or reduced leaflet mobility Paucity (restriction) of color Doppler transvalvular flow	Dense fixed hyperechogenic tissue Normal leaflet morphology Normal or reduced leaflet mobility	Presence of vegetation(s) Valve leaflet thickening Leaflet abnormality (torn/avulsed/perforated) Paravalvular complications (abscess, pseudoaneurysm, fistula)
CCTA	Leaflet calcificationHyperdense leaflet thickening RELM	No leaflet calcification Normal leaflet morphology and mobility	No leaflet calcification HALT RELM HAM	Possible leaflet calcification Hypodense semicircular or circular structure along and beneath the valve ring/stent High attenuation values (>200 HU)	No leaflet calcification Paravalvular complications (vegetations, abscess, pseudoaneurysm, fistula)
CMR	Reduced anatomic orifice area evidenced by cine CMR and planimetry Reduced EOA in phase-contrast CMR Abnormal aortic regurgitant volume and fraction in phase-contrast CMR	Reduced indexed anatomic orifice area evidenced by cine CMR and planimetry	No leaflet calcification Reduced anatomic orifice area evidenced by cine CMR and planimetry Abnormal aortic regurgitant volume and fraction in phase-contrast CMR	Reduced anatomic orifice area evidenced by cine CMR and planimetry Possible abnormal aortic regurgitant volume and fraction in phase-contrast CMR	Reduced anatomic orifice area evidenced by cine CMR and planimetry Abnormal aortic regurgitant volume and fraction in phase-contrast CMR
Nuclear imaging					
18-NaF PET/CT	Increased 18F-NaF uptake at the level of the bioprosthetic valve leaflets	No 18F-NaF uptake at the level of the bioprosthetic valve leaflets	Increased 18F-NaF uptake at the level of the bioprosthetic valve leaflets	No available data	Increased 18F-NaF uptake at the level of the bioprosthetic valve leaflets
18F-FDG PET/CT	No 18F-FDG uptake at the level of the bioprosthetic valve or paravalvular region	No 18F-FDG uptake at the level of the bioprosthetic valve or paravalvular region	No available data	No available data	Increased 18F-FDG uptake at the level of the bioprosthetic valve and paravalvular region

Abbreviations: 18F-FDG = 18F-fluorodeoxyglucose; 18-NaF-18F = sodium fluoride; CCTA = cardiac computed tomography angiography; HALT = hypoattenuated leaflet thickening; HAM = hypoattenuation affecting motion; PET = positron emission tomography; RELM = reduced leaflet motion; TEE = transesophageal echocardiography; TTE = transthoracic echocardiography.

## Data Availability

Not applicable.

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
