# Peer review of "Imaging of Bioprosthetic Valve Dysfunction after Transcatheter Aortic Valve Implantation"

_diagnostics, 2023, doi:10.3390/diagnostics13111908_

Round 1
Reviewer 1 Report
The current review paper on the use of different imaging modalities to evaluate aortic valve prostheses after TAVI is very well written and thoroughly analyzes each aspect of prosthetic dysfunction and the proper method to evaluate each aspect of the matter. I have no corrections or anything to add.
The use of the English language is fine.
I can not see the text in the tables very well (maybe a formatting problem?). I would also like to see a central figure in the paper to summarize the main concept of the review (maybe some examples of multimodality imaging in this area).
Author Response
The current review paper on the use of different imaging modalities to evaluate aortic valve prostheses after TAVI is very well written and thoroughly analyzes each aspect of prosthetic dysfunction and the proper method to evaluate each aspect of the matter. I have no corrections or anything to add.
Thank you for your benevolent comment.
I can not see the text in the tables very well (maybe a formatting problem?). I would also like to see a central figure in the paper to summarize the main concept of the review (maybe some examples of multimodality imaging in this area).
Thank you for pointing out the formatting error. We did correct the formatting and we added beside the established central figure (i.e. figure 1) (multimodality imaging cases) also 4 other Figures of case examples as well as a flow-chart figure depicting the different clinical scenarios.
Reviewer 2 Report
Transcatheter aortic valve replacement is widely used in the clinical practice with improved and reassuring results in presence of prolonged follow-up; otherwise, it is necessary to underline that the acronym is a misnomer because the native aortic valve is not replaced but rather displaced and splinted against the wall of the aorta at the time of bioprosthetic valve insertion. The procedure of TAVI is rapidly gaining a wide implementation as an easy procedure and a treatment option in younger low-risk population rather than in the previous experiences.
It has been previously suggested that the impact of repeated valve closure and trauma can be fundamental as a causative element in aortic stenosis. Therefore, patients with TAVI present a unique opportunity to investigate the pathophysiology of aortic stenosis in the absence of the ongoing cyclic mechanical trauma of valve closure. This is a very interesting issue in the setting of the imaging studies of the patients who underwent to TAVI procedure.
This technical aspect can be take into account when we decide to study with different imaging techniques the post-TAVI patients .
Therefore, in these patients treated with TAVI it is also important to investagate if an active regulated pathobiological process continues in the native aortic valve, despite valve immobilization .
As a consequence, recent clinical studies showed that a contemporary degenerative process is incoming in the native aortic valve as well as in bioprostetic valve.
It has been previously suggested that the impact of repeated valve closure and trauma is fundamental to aortic stenosis.
The long-term durability of transcatheter aortic valves is yet to be clearly established, we aimed to establish whether bioprosthetic valve durability or degeneration was appreciably different between patients with TAVI or SAVR at midterm follow-up.
These are in my opinion the challlenging questions to be furher explored.
The powerful prediction of valve dysfunction provided by 18F-NaF in both bioprosthetic SAVR and TAVI valves, our data set provides a unique opportunity to compare early valve degeneration in age-matched bioprosthetic SAVR and TAVI valves, thereby helping address one of the most important current questions.
Aortic valve disease after TAVI procedure demonstrates an intrinsic activity, suggesting that aortic stenosis is an active disease process that is independent of motion and mechanical injury. In some studies, across 3 complementary and distinct imaging modalities, TAVI degeneration appears to be of a similar magnitude to bioprosthetic SAVR.
These issues are of course the outstanding aspects in the evaluation and thew assessment of bioprostetic valve dysfunction after TAVI
This paper represents a current and enough appropriate paper of rewiew on this topic .
After a brief but exhaustive introduction, the authors introduce a very articulated and complex first table, in our opinion not easy to acquire, with the preliminary introduction of the criteria attributable to the "Red flags". This first table introduces the criteria: diagnostic, staging and progression of the disease. A second table also included in the second chapter proposes the comparison between the different imaging modalities in the evaluation and quantification of the dysfunction of the bioprosthesis.
In our opinion, the tables are unclear and intelligible and must certainly be reformulated. The 3rd chapter is of particular interest and originality, addressing the issues of paravalvular regurgitation and patient prosthesis mismatch.
However , the discussions of the themes appear too verbose in this case and devoid of any iconographic support . This is the aspect that is most lacking and which is, so to speak, in contrast with the layout of the paper which could imply an important iconographic contribution. Again, chapter seven which deals with the issue of early diagnosis of bioprosthetic valve degeneration is certainly one of the most current and deserves a discussion more related to the clinical impact. Finally, the conclusions appear strictly limited and concise as if they were the conclusions of an abstract.
Transcatheter aortic valve replacement is widely used in the clinical practice with improved and reassuring results in presence of prolonged follow-up; otherwise, it is necessary to underline that the acronym is a misnomer because the native aortic valve is not replaced but rather displaced and splinted against the wall of the aorta at the time of bioprosthetic valve insertion. The procedure of TAVI is rapidly gaining a wide implementation as an easy procedure and a treatment option in younger low-risk population rather than in the previous experiences.
It has been previously suggested that the impact of repeated valve closure and trauma can be fundamental as a causative element in aortic stenosis. Therefore, patients with TAVI present a unique opportunity to investigate the pathophysiology of aortic stenosis in the absence of the ongoing cyclic mechanical trauma of valve closure. This is a very interesting issue in the setting of the imaging studies of the patients who underwent to TAVI procedure.
This technical aspect can be take into account when we decide to study with different imaging techniques the post-TAVI patients .
Therefore, in these patients treated with TAVI it is also important to investagate if an active regulated pathobiological process continues in the native aortic valve, despite valve immobilization .
As a consequence, recent clinical studies showed that a contemporary degenerative process is incoming in the native aortic valve as well as in bioprostetic valve.
It has been previously suggested that the impact of repeated valve closure and trauma is fundamental to aortic stenosis.
The long-term durability of transcatheter aortic valves is yet to be clearly established, we aimed to establish whether bioprosthetic valve durability or degeneration was appreciably different between patients with TAVI or SAVR at midterm follow-up.
These are in my opinion the challlenging questions to be furher explored.
The powerful prediction of valve dysfunction provided by 18F-NaF in both bioprosthetic SAVR and TAVI valves, our data set provides a unique opportunity to compare early valve degeneration in age-matched bioprosthetic SAVR and TAVI valves, thereby helping address one of the most important current questions.
Aortic valve disease after TAVI procedure demonstrates an intrinsic activity, suggesting that aortic stenosis is an active disease process that is independent of motion and mechanical injury. In some studies, across 3 complementary and distinct imaging modalities, TAVI degeneration appears to be of a similar magnitude to bioprosthetic SAVR.
These issues are of course the outstanding aspects in the evaluation and thew assessment of bioprostetic valve dysfunction after TAVI
This paper represents a current and enough appropriate paper of rewiew on this topic .
After a brief but exhaustive introduction, the authors introduce a very articulated and complex first table, in our opinion not easy to acquire, with the preliminary introduction of the criteria attributable to the "Red flags". This first table introduces the criteria: diagnostic, staging and progression of the disease. A second table also included in the second chapter proposes the comparison between the different imaging modalities in the evaluation and quantification of the dysfunction of the bioprosthesis.
In our opinion, the tables are unclear and intelligible and must certainly be reformulated. The 3rd chapter is of particular interest and originality, addressing the issues of paravalvular regurgitation and patient prosthesis mismatch.
However , the discussions of the themes appear too verbose in this case and devoid of any iconographic support . This is the aspect that is most lacking and which is, so to speak, in contrast with the layout of the paper which could imply an important iconographic contribution. Again, chapter seven which deals with the issue of early diagnosis of bioprosthetic valve degeneration is certainly one of the most current and deserves a discussion more related to the clinical impact. Finally, the conclusions appear strictly limited and concise as if they were the conclusions of an abstract.
So in my opinion the paper proposed as a review on a very interesting and challenging issues need a moderate revision focusing an effective relationship between the tables (must be revised) and the text; the conclusion must be revised and astill integration with images can enrich the paper on the clinical point of view.
Author Response
After a brief but exhaustive introduction, the authors introduce a very articulated and complex first table, in our opinion not easy to acquire, with the preliminary introduction of the criteria attributable to the "Red flags". This first table introduces the criteria: diagnostic, staging and progression of the disease. A second table also included in the second chapter proposes the comparison between the different imaging modalities in the evaluation and quantification of the dysfunction of the bioprosthesis.
In our opinion, the tables are unclear and intelligible and must certainly be reformulated.
Thank you for giving us the possibility to improve our manuscript. Regarding the tables, we have now simplified it by dividing table 1 into 2 separate tables. Further, we have improved table 2 by visualization with different colors and also by removing unnecessary information from the table.
However, the discussions of the themes appear too verbose in this case and devoid of any iconographic support . This is the aspect that is most lacking and which is, so to speak, in contrast with the layout of the paper which could imply an important iconographic contribution. Again, chapter seven which deals with the issue of early diagnosis of bioprosthetic valve degeneration is certainly one of the most current and deserves a discussion more related to the clinical impact.
Thank you for pointing this out. We have added more information in chapter seven and have improved now the section significantly.
Finally, the conclusions appear strictly limited and concise as if they were the conclusions of an abstract
We have now adapted the conclusion in a more comprehensive way that readers understand that echocardiography is the first line imaging modality in suspected evaluations of structural and functional valve degeneration. Multimodality imaging allows a more personalized approach in selected cases and provides a window into the pathophysiology of bioprosthetic valve failure.
Reviewer 3 Report
Thank you for asking me to review this manuscript This is undoubtedly an interesting review, worthy of attention. The manuscript is well written. Surely further novel modalities and future improvements will add something new about this topic.
Author Response
Thank you for asking me to review this manuscript This is undoubtedly an interesting review, worthy of attention. The manuscript is well written. Surely further novel modalities and future improvements will add something new about this topic.
Thank you for the benevolent comment.
Reviewer 4 Report
Thank you for the opportunity to review this manuscript.
The authors discuss the increasing use of transcatheter aortic valve implantation (TAVI) in younger and lower-risk populations and the need to investigate the long-term durability of these bioprosthetic aortic valves in the near future. However, diagnosing dysfunction of these valves is challenging. The text focuses on the contemporary and future roles, advantages, and limitations of multi-modality imaging to monitor the integrity of transcatheter heart valves.
The manuscript is written very concisely, like a summary of a guideline from beginning to end. However, some concepts need to be expanded more in-depth and should be illustrated to help lighten the text and make it easier for the reader to grasp, especially when new imaging concepts are introduced.
-
The authors state that echocardiography (TTE and TEE) is an important first-line diagnostic. However, it would be important not only to write that TTE is dependent on the acoustic window but also TEE. L84-85: could the authors elaborate on the need for different esophageal or trans-gastric views needed due to prosthesis artifacts to better characterize leaflet and sub-valvular morphology? In L156-157: the authors write about the quantification of PVR, could they give a figure on how to differentiate trans- from para-prosthetic regurgitation in a more convenient manner? A flowchart would be appropriate. What is the role of 3D-Color-TEE in the diagnosis?
-
The authors rightly state the role of CMR in the quantification of regurgitation. However, the reader would be interested in a figurative illustration of how to avoid THV stent artifacts during the acquisition of CMR sequences.
-
The authors describe the method of “videodensitometry” in L179 and Table. Since this method is not represented in the guidelines, the authors should give a figurative illustration of this potential method to increase comprehensibility for the reader.
-
The authors emphasize the importance of the distinction between SVD and PPM. However, a concise flowchart would be ideal to convey the very complex diagnostic “journey” with several parameters and stress-echocardiography of the diagnosis of PPM.
-
The authors describe the importance of HALT-diagnosis in early stages to prevent further eventual deterioration of TAVI prostheses. Could the authors give an example from CCTA and Echo? This might increase visualization and comprehensibility for the reader.
-
The authors elaborate on the novel method of 18F-NaF-PET in prosthesis calcification. However, they should concisely explain the underlying molecular mechanism, why it works for valve calcification, and not only as a tracer for bone turnover, for the less knowledgeable reader.
-
The authors should describe and discuss what their manuscript offers in comparison to Pibarot et al., J Am Coll Cardiol. 2022 Aug, 80 (5) 545–561, since the reference to this important state of the art review is missing.
-
The authors should more figuratively describe the role of ARI in L139-142, i.e., explain better what is meant with inversely correlated.
-
The authors write in several passages about the potential role of multi-modal imaging but give no references. This review should give the reader the opportunity to find important references for self-study and clinical implementation, like in L75-76, and L106-108: is there a reference of PET for SVD but not endocarditis, and so on.
-
Table 2 is not readable due to formatting issues.
-
Missing Figure 1.
-
Table 1: abbreviations in legend explained but not used in the main part of the table, like HALT.
Author Response
The authors state that echocardiography (TTE and TEE) is an important first-line diagnostic. However, it would be important not only to write that TTE is dependent on the acoustic window but also TEE. L84-85: could the authors elaborate on the need for different esophageal or trans-gastric views needed due to prosthesis artifacts to better characterize leaflet and sub-valvular morphology?
Thank you for this important input. We elaborated the importance of TEE and also its dependency on the acoustic window and the need of a deep transgastric view that mimic an apical 5 chamber and allows to better evaluate the leaflet motion, prosthetic aortic regurgitation and sub-valvular morphology. Moreover we added new figures reflecting the demonstration of paravalvular regurgitation after TAVI assessed by TTE/TEE.
What is the role of 3D-Color-TEE in the diagnosis?
This is a very interesting point. Thank you. We have now added information on the role of 3D TTE-TEE and its ability to for quantify PVR and measure the vena contracta area. TEE can provide more detailed assessment of the aortic regurgitation and aortic valve morphology and function. 3D echocardiography can reliably identify especially the mechanism of aortic regurgitation and with the complementary use of 3D color mode provides important diagnostic information about the regurgitant jets size, location, and severity.
The authors rightly state the role of CMR in the quantification of regurgitation. However, the reader would be interested in a figurative illustration of how to avoid THV stent artifacts during the acquisition of CMR sequences.
Thank you for pointing this out. As this review is intended for the general cardiologist, we fell that providing details into CMR acquisition would be beyond the scope of the current review. However to address the reviewer comment, we have added a short information within the text, pointing out, that assessment more distal to the stent or in the aorta descendens might be preferred to overcome the issue of artifact.
The authors describe the method of “videodensitometry” in L179 and Table. Since this method is not represented in the guidelines, the authors should give a figurative illustration of this potential method to increase comprehensibility for the reader
Thank you for pointing this out. As “videodensitometry” is indeed not recommended within the guidelines and not to confuse the reader with research techniques, and for the sake of the extensive manuscript, we have decided to eliminate this section.
The authors emphasize the importance of the distinction between SVD and PPM. However, a concise flowchart would be ideal to convey the very complex diagnostic “journey” with several parameters and stress-echocardiography of the diagnosis of PPM.
This is an excellent proposition. We have now added a flowchart, which showed the differentiation between SVD, PPM and thrombosis. We highlighted the fact that low dose dobutamine stress echo may play indeed an important role in PPM assessment.
The authors describe the importance of HALT-diagnosis in early stages to prevent further eventual deterioration of TAVI prostheses. Could the authors give an example from CCTA and Echo? This might increase visualization and comprehensibility for the reader.
Thank you for this valuable comment. We have now added new figures to depict the role of echocardiography and CT in imaging thrombosis and HALT after TAVI.
The authors elaborate on the novel method of 18F-NaF-PET in prosthesis calcification. However, they should concisely explain the underlying molecular mechanism, why it works for valve calcification, and not only as a tracer for bone turnover, for the less knowledgeable reader.
We explained the underlying molecular mechanism which explain 18F-NaF PET as a marker of vascular calcification in AS and atherosclerosis affecting the aorta, coronary and carotid arteries. This explanation was supported by the autoradiography and immunohistochemical data from Dweck et al demonstrating a good correlation between 18F-NaF activity and osteocalcin staining: a well-recognized osteogenic protein that itself binds to hydroxyapatite (Ref.: Dweck MR, Jenkins WS, Vesey AT, Pringle MA, Chin CW, Malley TS, et al. 18F-sodium fluoride uptake is a marker of active calcification and disease progression in patients with aortic stenosis. Circulation: Cardiovascular Imaging. 2014;7(2):371-8).
The authors should describe and discuss what their manuscript offers in comparison to Pibarot et al., J Am Coll Cardiol. 2022 Aug, 80 (5) 545–561, since the reference to this important state of the art review is missing.
Our Review, in comparison to the review from Pibarot et al., bring a detailed comparison between the two official classification of structural valve deterioration with ESC and VIVID. In addition in this review, we presented the contemporary role, advantages, and limitations of TTE and TEE, CCTA, CMR and PET to monitor anatomical and functional THV integrity and their potential impact on patient management. We developed the fact that CMR could have a significant role after TAVI in the assessment of intravalvular and/or paravalvular regurgitation severity, providing a superior prognostic value than echocardiography. Moreover we evaluated the future directions concerning an early diagnosis of bioprosthetic valve deterioration (BVD). In Contrast the state of art Review from Pibarot et al gave a detailed definition of bioprosthetic valve dysfunction following aortic and mitral valve replacement and measurement methods of non structural and structural BVD after aortic or mitral biological valve replacement. Instead, we presented a detailed definition of BVD, non structural-BVD, valve thrombosis, endocarditis and their common end stage BVF.
The authors should more figuratively describe the role of ARI in L139-142, i.e., explain better what is meant with inversely correlated.
The ARI and the severity of the PVR are inversely proportional; ARI <25% is associated with increased 1-year mortality risk in comparison with patients with ARI >25%. In order to evaluate PVL after TAVI and to identify patients who will benefit from corrective measures (post-dilatation, valve-in-valve implantation), Sinning et al. have suggested using a multimodal approach combining hemodynamic measurements and imaging modalities (aortography and TTE/TEE). When there is no PVL and/or an ARI >25% are present, no additional measures need to be taken. While in patients with more than-mild PVL and/or ARI <25%, the evaluation of PVL using TTE/TEE is recommended to elucidate the cause of PVL.
The authors write in several passages about the potential role of multi-modal imaging but give no references. This review should give the reader the opportunity to find important references for self-study and clinical implementation, like in L75-76, and L106-108: is there a reference of PET for SVD but not endocarditis, and so on.
Regarding the role of PET imaging in SVD, we added the references of Tzolos et al., (Nuclear Cardiology and Multimodal Cardiovascular Imaging, E-Book: A Companion to Braunwald's Heart Disease. 2021:385.), Dweck et al. (Circulation: Cardiovascular Imaging. 2014;7(2):371-8.) and Irkle et al. (Nature communications. 2015;6(1):1-11.) combining advanced techniques which permit to detect early stage of valve degeneration and may allow personalized and improved management strategies for THV patients.
Table 2 is not readable due to formatting issues.
Thank you. We have corrected this error.
Missing Figure 1.
Figure 1 is the graphical Abstract
Table 1: abbreviations in legend explained but not used in the main part of the table, like HALT.
Thank you for pointing out this error. We have corrected it.
Reviewer 5 Report
In this article, the authors focus on different imaging modalities to monitor the integrity of transcatheter heart valves. The article is divided in paragraphs according to the type of valve dysfunction (Structural valve deterioration SVD, non-SVD, para valvular regurgitation, patient-prosthesis mismatch, valve thrombosis, infective endocarditis and bioprosthetic valve failure). For each of these, the authors describe the characteristics and discuss the role of different imaging modalities in diagnosis and follow up.
I think that the article is well written and interesting to read, even if it does not add much to our knowledge of the subject. Anyway, as you state really well in your introduction, TAVI is increasingly performed in younger intermediate and lower-risk populations, and consequently more efforts should be done to investigate long‐term durability of bioprosthetic aortic valves.
As a remark, pleas check Table 2 of the paper because the first letter of any paragraph is missing
Author Response
As a remark, pleas check Table 2 of the paper because the first letter of any paragraph is missing.
Thank you for pointing out this error. We have corrected this issue.
Round 2
Reviewer 4 Report
The authors have done a great job, the manuscript text has clearly gained in structure and comprehensibility, and the work also encourages to delve further into this very tricky topic in self-study. Thank you very much. Nevertheless, the figures 2-5 are unfortunately not to be seen in the manuscript, this would have interested me of course personally very much.